# Protective Effect of Acteoside on Ovariectomy-Induced Bone Loss in Mice

**DOI:** 10.3390/ijms20122974

**Published:** 2019-06-18

**Authors:** Lingling Yang, Bo Zhang, Jingjing Liu, Yanhong Dong, Yanting Li, Nan Li, Xiaojun Zhao, Hunter Snooks, Changling Hu, Xueqin Ma

**Affiliations:** 1Department of Pharmaceutical Analysis, School of Pharmacy, Key Laboratory of Hui Ethnic Medicine Modernization, Ministry of Education, Ningxia Medical University, 1160 Shenli Street, Yinchuan 750004, China; 15926361499@163.com (L.Y.); zhangbobo0624@163.com (B.Z.); 15709604029@163.com (J.L.); dyh794200808@163.com (Y.D.); liyanting134@163.com (Y.L.); linan950618@163.com (N.L.); zhaoxiaojunwr@163.com (X.Z.); 2Laboratory for Functional Foods and Human Health, Center for Excellence in Post Harvest Technologies, North Caroline A & T State University, North Carolina Research Campus, 500 Laureate Way, Kannapolis, NC 28081, USA; hdsnooks@ncat.edu (H.S.); chu@ncat.edu (C.H.)

**Keywords:** acteoside, ovariectomized mice, anti-osteoporotic, RANKL, RANK, TRAF6

## Abstract

Acteoside, an active phenylethanoid glycoside compound isolated from herbs of *Cistanche*, was chosen for the investigation of anti-osteoporotic effect on postmenopausal osteoporosis by using an ovariectomized (OVX) mice model. The results from in vivo experiments showed that after daily oral administration of acteoside (20, 40, and 80 mg/kg body weight/day) for 12 weeks, bone mineral density and bone biomechanical properties of OVX mice were greatly enhanced, with significant improvement in bone microarchitecture. Furthermore, biochemical parameters of bone resorption markers as well as bone formation index, including tartrate-resistant acid phosphatase, cathepsin K, deoxypyridinoline, alkaline phosphatase, and bone gla-protein, were ameliorated by acteoside treatment, whereas the body, uterus, and vagina wet weights were seemingly not impacted by acteoside administration. Acteoside significantly affected osteoclastogenesis by attenuating nuclear factor kappa B (NF-κB) and stimulating phosphoinositide 3-kinase (PI3K)/protein kinase B (AKT) signal pathways through down-regulated levels of tumor-necrosis factor receptor-associated factor 6 (TRAF6), receptor activator of nuclear factor kappa B ligand (RANKL), RANK, NFKBIA, IκB kinase β, nuclear factor of activated T-cells c2 (NFAT2), and up-regulated expressions of PI3K, AKT, and c-Fos. Accordingly, the current research validated our hypothesis that acteoside possesses potent anti-osteoporotic properties and may be a promising agent for the prevention of osteoporosis in the future.

## 1. Introduction

Osteoporosis is a severe health threat defined as a systemic skeletal disease and characterized by low bone mass and deterioration of trabecular bone structure, thereby being a significant risk factor for bone fracture [1]. Osteoporosis is becoming more prevalent as the elderly population is increasing worldwide, and it is estimated that 50% of women and 20% of men aged 50 years and over will suffer one or more fractures mainly caused by osteoporosis during their remaining life, thus posing a great challenge for the prevention and treatment of osteoporosis [2]. Although osteoporosis has received more attention lately and synthetic drugs used to treat this disease are indeed effective, various kinds of adverse effects which may be related to these agents make them far from ideal [3]. Therefore, more attention has been given to natural medicinal plants, especially some edible herbs and isolated active compounds, to treat this aliment. A series of medicinal herbs were discovered to possess anti-osteoporotic effects in the laboratory, which include *Epimedium brevicornu*, *Ligustrum lucidum*, *Cistanche deserticola*, etc. [4]. Unfortunately, the responsible bioactive components of these herbs, as well as the underlying molecular mechanisms, are still not clear.

It is well known that bones are constantly undergoing a process of remodeling, including the formation of bone matrices and the resorption of bone, which are guided by osteoblasts and osteoclasts, respectively [5]. It is widely believed that the bone remodeling process is mainly dependent on the vitality and differentiation of osteoclasts [6]. The osteoclast is a unique myeloid-derived multinucleated giant cell, and colony-stimulating factors (M-CSF) as well as receptor activators of nuclear factor-κB (NF-κB) ligand (RANKL) are regarded as critical cytokines for the formation and activation of osteoclasts [7]. It was proven that osteoclastogenesis is activated when the receptor activator of NF-κB (RANK) interacts with its ligand RANKL, as well as when tumor-necrosis factor (TNF) receptor-associated factor 6 (TRAF6) is combined with this complex [8]. Consequently, the binding of TRAF6/RANKL/RANK activates a series of downstream signaling pathways including NF-κB and phosphatidylinositol 3-kinase AKT (PI3K/AKT) [9], and ultimately stimulates the activation of c-Fos and nuclear factor of activated T-cells c2 (NFAT2) [10]. 

Acteoside, also called verbascoside, is one of the representative phenylethanoid glycoside (PhG) components existing in dozens of medicinal plants, including the herbs of genus *Verbascum* and *Cistanche*. In recent years, acteoside has garnered more attention due to its excellent biological activities including antioxidant [11], anti-inflammatory [12], neuronal protective [13], and improved muscle atrophy properties [14]. According to the record of the Chinese pharmacopeia, *C. deserticola* is an edible traditional Chinese medicine (TCM) which was traditionally used to treat kidney deficiency symptoms including muscle weakness and lumbar debility, and PhGs including echinacoside and acteoside were established as the main bioactive constituents in this herb [15]. According to the TCM theory of “kidney dominates bone”, TCM possessing activities of nourishing and strengthening the kidney may be employed as anti-osteoporotic agents in clinical. In addition, published experiments using both osteoblasts in vitro and ovariectomized (OVX) mice in vivo proved the anti-osteoporotic effect of *C. deserticola* [16,17,18,19,20], and acteoside was also reported to reduce bone loss by blocking osteoclast activation [21]. All the above evidence concurs with our hypothesis that acteoside may have bone protective and bone reparative properties. Therefore, the objective of this study is to determine the therapeutic effect of acteoside on OVX mice in vivo as well as the underlying molecular mechanism related to TRAF6/RANKL/RANK-mediated NF-κB and PI3K/AKT signals.

## 2. Results

### 2.1. Body, Uterine, and Vaginal Weights

To test whether acteoside possessed the phytoestrogen characteristic or not, the body, uterine, and vaginal weights of animals were estimated. Six groups of mice exhibited similar initial mean body weights, as seen in Figure 1. However, despite having similar food intake in all groups regardless of intervention, the mice in the OVX model group weighed significantly more than the SHAM mice (*p* < 0.01) in the sixth week, an effect that lasted to the end of the experiment after post-ovariectomy, when weight had increased by 15.07% and 21.47%, respectively (*p* < 0.05). These increased body weights of the OVX mice were not reversed by any of the three doses of acteoside administration (AceL, AceM, and AceH as low-, moderate-, and high-dosage groups, where the dosages were 20, 40, and 80 mg/kg/day, respectively). In contrast to the body weights, surgery of ovariectomy caused significant atrophy of uterine and vaginal tissue in OVX model mice by 76.98% and 73.28% (*p* < 0.001), respectively, as compared to SHAM mice. Similarly, acteoside did not elicit any effect, whereas administering estradiol valerate (EV) significantly recovered the uterine and vaginal wet weights as compared to the mice of OVX model group (*p* < 0.01). In addition, the tissue weights of heart, liver, spleen, lung, and kidney exhibited no statistically significant change in each group. 

### 2.2. Acteoside Improved the Bone Biomechanical Properties

To estimate whether acteoside could improve bone strength and elasticity or not, a three-point bending test was employed. The levels of maximal load and stiffness, as seen in Figure 2, were significantly decreased by 29.56% and 26.13% in the OVX model group as compared to the SHAM group (*p* < 0.01) after 12 weeks of ovariectomy operation, respectively. When administrated with acteoside, the levels of maximal load and stiffness displayed a significant dose-dependent increasing trend, and the high and moderate dosages of acteoside could improve the maximal load and stiffness by 31.42%, 23.09%, 32.67%, and 18.28%, respectively, when compared to the OVX model mice.

### 2.3. Acteoside Prevented Bone Loss in Ovariectomized Mice

To estimate the microarchitecture of sub-chondral trabecular bones of OVX mice, the technology of micro-CT was employed. As seen in Figure 3, a series of significantly decreased trends of bone mineral density (BMD), bone mineral content (BMC), tissue mineral content (TMC), bone volume fraction (BVF), trabecular thickness (Tb.Th), and trabecular number (Tb.N) by 68.23%, 77.81%, 92.31%, 89.61%, 62.72%, and 79.81% were obtained in OVX model mice, respectively, as compared to the SHAM-operated groups, and an increased trend of trabecular separation (Tb.Sp) by 85.40% was also observed in mice of the OVX model group. However, OVX mice treated with acteoside dose-dependently increased BMD (50.69–53.29%), BMC (61.27–69.31%), TMC (86.22–86.82%), BVF (83.33–86.21%), Tb.Th (33.33–52.20%), and Tb.N (73.22–82.98%), and decreased Tb.Sp (78.12–87.50%) as compared to the mice of OVX model group. 

### 2.4. Acteoside Promoted Bone Formation and Inhibited Resorption Markers

Some specific markers, including alkaline phosphatase (ALP), bone gla-protein (BGP), tartrate-resistant acid phosphatase (TRAP), cathepsin K, and deoxypyridinoline (DPD), were often used to reflect the extent of bone formation and bone resorption (Figure 4). Concerning the ALP and BGP, two bone formation markers, a rising trend (but not a statistically significant change) in ALP activity was observed in all the acteoside treated groups as compared to the OVX group; acteoside at high (80 mg/kg) and low (20 mg/kg) dosages significantly improved the BGP levels in the OVX mice (*p* < 0.05). For the resorption markers, the ovariectomy operation resulted in significant enhancement of TRAP, cathepsin K, and DPD activities by 44.14%, 36.19%, and 51.68%, respectively, as compared to the SHAM group. It was noteworthy that the activities of all these bone resorption markers in the OVX mice were notably suppressed after being treated with acteoside. Quantitatively, the high dosage (80 mg/kg/day) of acteoside restrained the levels of TRAP and DPD by 50.71% and 60.48% (*p* < 0.001), respectively, as compared to the OVX model group; furthermore, the medium dosage (40 mg/kg/day) of acteoside suppressed the activity of cathepsin K by 45.30% (*p* < 0.001) when compared to the OVX model group. 

### 2.5. Acteoside Influenced the Protein Expression Levels of TRAF6, RANKL, RANK, NF-κB, PI3K, AKT, IKKβ, NFAT2, and C-Fos

To clear the anti-osteoporotic molecular mechanisms of acteoside, we performed the corresponding western blot experiments. As expected, acteoside exhibited no cytotoxic effects on RAW 264.7 induced osteoclast cells (10^−5^, 10^−6^ and 10^−7^ mol/L, respectively) according to the 3-(4,5-Dimethylthiazol−2-yl)-2,5-diphenyltetrazolium bromide (MTT) assay results (Figure 5). When treated with acteoside, the protein levels of TRAF6, RANKL, RANK, NF-κB, IKKβ, and NFAT2 were significantly down-regulated, while the expressions of PI3K, AKT, and c-Fos were up-regulated as compared to the control (Figure 6). A hypothesized molecular mechanism of acteoside acting on osteoclastogenesis was depicted in Figure 7. Briefly, acteoside inhibited the binding of RANKL, RANK, and TRAF6, followed by the downstream signals, including NF-κB pathway, being suppressed and the PI3K/AKT signal being activated. Then, the activities of c-Fos and NFAT2 were stimulated, and ultimately osteoclastogenesis was suppressed. 

## 3. Discussion

There is a growing recognition and approval of TCM for the treatment of complicated aliments based on extensive experience which has been accumulated over thousands of years in China. More remarkably, TCM theory points out that the bone development system is governed and dominated by the kidney, which means the kidney is an essential organ for nourishing bone development during growth [22]; thus TCM that possesses properties for invigorating the kidney can be used as an anti-osteoporotic agent for clinical purposes. *C. deserticola*, a traditional TCM and a common natural health food, has long been used as a tonic agent in China and Japan. *C. deserticola* is also called “desert ginseng” due to its traditional uses of nourishing the kidney as well as its excellent curative effects including anti-oxidative, anti-nociceptive, anti-inflammatory, anti-fatigue, and neuroprotective activity [16]. Furthermore, previous studies have discovered that *C. deserticola* dose-dependently improved bone quality in animals with osteoporosis induced by ovariectomy. Based on the TCM theory of “kidney dominates bone” as well as published data, *C. deserticola* indeed possessed potent anti-osteoporotic activity; however, the corresponding compounds and the underlying molecular mechanisms are still unclear. According to Chinese pharmacopeia, the compound of PhG including echinacoside and acteoside are the main and effective chemical compositions of *C. deserticola*, and acteoside is a well-studied natural component that is widely distributed in dozens of medicinal plants [23]. Given the detailed anti-osteoporosis mechanisms of *C. deserticola*, including acteoside, are still unclear, the present study was designed to systematically evaluate the therapeutic effect of acteoside as well as underlying molecular mechanisms. As expected, our research demonstrated that acteoside could enhance bone mineral density and biomechanical properties as well as improve the microarchitecture of bone trabecular, and finally prevent bone loss in osteoporotic mice induced by ovariectomy. 

There is no doubt that estrogen plays an important role in stimulating bone metabolism, and postmenopausal osteoporosis, a sex-steroid deficiency state characterized by decreased bone mineral density and increased risk of fracture, is believed to be mainly caused by estrogen deficiency [24]. According to previous data from the literature, after OVX surgery and during the process of postmenopausal osteoporosis, uterine and vaginal weights of OVX animals were significantly decreased, whereas the body weight was obviously increased [25]. In the present study, as expected, postmenopausal osteoporosis caused an excess of whole-body weight, and sharp decline of uterine and vaginal weights of OVX mice. When complemented with positive drug EV, an estrogen agent that has been proved effective in the treatment of postmenopausal osteoporosis in clinical conditions, the gained body weight and atrophied uterine and vaginal weights were significantly reversed. In contrast, acteoside did not prevent the induced body weight gain as well as uterine and vaginal weight loss in the OVX mice, which implied that the stimulation of unwanted proliferation of body growth and atrophy of uterine and vaginal tissues in the OVX mice were not influenced by acteoside, but reversed by EV treatment. All the above results indicate that acteoside improved the bone microarchitecture and biomechanics through another mechanism rather than the phytoestrogen characteristic such as EV. 

As well as BMD as one of the major markers for the diagnosis of osteoporosis [25], micro-CT and biochemical detection were also used to provide additional direct information of trabecular microarchitecture to assess bone fragility [26]. The micro-CT technology, instead of histological staining, can give us both the intuitive and quantitative data of the microarchitecture of trabecular bone, and the 3D image of trabecular bone which supplied by micro-CT method was very intuitive. The area of the trabecular bone can be observed, and quantitative data including BVF, BMD, BMC, TMC, TMD, Tb.N, Tb.Th, Tb. Sp, etc. were also obtained. Therefore, the data obtained from the micro-CT and biochemical detection were believed to be more meaningful and comprehensive for the diagnosis of osteoporosis. In the present study, acteoside dose-dependently (20, 40, and 80 mg/kg/day) improved the BMD in OVX mice, and most of the variables in the micro-CT experiment were significantly ameliorated, including the reduction of BMC, TMC, BVF, Tb.N, and Tb.Th in OVX mice, which was consistent with the published data [27], whereas the increment of Tb.Sp was totally reversed; moreover, in parallel with the micro-CT findings, the biomechanical parameters, including the maximal load and stiffness, were also significantly enhanced by acteoside treatment. Therefore, our data implied the therapeutic effect of acteoside on bone microarchitecture of OVX mice. 

During the process of postmenopausal osteoporosis, the reduction of BMD and deterioration of the bone microarchitecture in the femur were always accompanied by significant changes in the bone remodeling markers including ALP, BGP, TRAP, cathepsin K and DPD [28]. It is well known that TRAP, cathepsin K, and DPD are regarded as typical markers for bone resorption, while ALP and BGP are widely considered to be bone formation indexes [29]. Treatment with acteoside prevented any significant increase in TRAP, cathepsin K, and DPD activities, and the level of BGP was also up-regulated in OVX mice, which means acteoside could both suppress bone resorption and enhance bone formation, and thus exhibited anti-osteoporotic properties. 

To the best of our knowledge, bone reconstruction and remodeling is a complicated physiological process that requires the function of osteoclasts [30]. It was confirmed that the activity and differentiation of osteoclasts were significantly increased during the menopause because of the sharply declined estrogen levels [31]. RANKL and its receptor, RANK, two key differentiation factors of osteoclast progenitors that belong to the TNF family [32], were believed to be able to stimulate osteoclast differentiation and then mediate osteoclastogenesis [33]. Furthermore, it was determined that the binding of RANKL to RANK could activate a variety of downstream signal pathways including NF-κB and PI3K/AKT, as well as some transcription factors including NFAT2 and c-Fos [34], and thus stimulated the development of osteoclasts and osteoclastogenesis. Nevertheless, RANK receptor protein lacked intrinsic enzymatic activity unless the TNF-receptor-associated factors (TRAFs) were recruited and were followed by the downstream signal cascades being transduced. TRAF6 was believed to be the primary adaptor molecule which was potentially useful for the function and differentiation of osteoclasts [5,32]. Hence, the RANKL/RANK/TRAF6 system was considered to be an important mechanism to explain osteoclastogenesis. In the present study, as seen in Figure 7, a hypothesized schematic mechanism by which acteoside suppressed osteoclastogenesis is depicted. Concisely, acteoside decreased RANKL, RANK, and TRAF6 expressions and therefore inhibited the system of RANKL/RANK/TRAF6, and consequently, the downstream pathway of NF-κB was inactivated and PI3K/AKT was activated, and ultimately, the level of NFAT2 decreased and the expression of c-Fos increased. Previous published data also showed that acteoside inhibited osteoclast differentiation by suppressing RANKL-induced activation of transcription factors such as NF-κB and NFATc1 [21], which is consistent with our results. Overall, the targeted modulation of the RANKL/RANK/TRAF6 system can be useful in the explanation of the therapeutic effects of acteoside on the bone metabolism in OVX mice. However, there are several limitations which should be considered in further studies, including the differences of bone cellular parameters and dynamic parameters in OVX mice, which should be detected using histomorphometric methods; it is more appropriate to induce postmenopausal osteoporosis in aged mice than young mice.

## 4. Materials and Methods

### 4.1. Preparation of Acteoside

Medicinal plants of *Cistanche deserticola* Y. C. Ma were obtained in September September 1, 2019 from Yongning county of Ningxia province, China, with the specific position of east longitude 106.026597 and northern latitude 38.262816. The stems were identified by Ling Dong (department of pharmacognosy, Ningxia Medical University) according to the standard Chinese Pharmacopeia procedures, and a corresponding specimen (#20150901) was deposited in the herbarium of the above department. For the isolation procedure, the dried and powdered stems of *C. deserticola* (30 kg) were reflux extracted for 2 h, 3 times, by using 70% ethanol as the solvent and the ratio of materials to solvents were set as 1:6 (*v*/*v*). Then, the filtrates were combined and purified by using macroporous resin (AB-8) column with a stepwise gradient ethanol–water solvent system (0%, 20%, 30%, 40%, 50%, and 60% ethanol, successively, each 60 L) to gain 6 fractions; and the fractions of 40% and 50% ethanol were collected and then subjected to further purification by using a sephadex column and semi-preparative liquid chromatography (LC) (Agilent 1100) to obtain a total of 20 g of acteoside. The structure was elucidated by detailed spectroscopic analyses, including nuclear magnetic resonance (NMR) (Bruker Avance 400 MHz) and MS spectra (Agilent 6540) data, and the purity was determined by applying the area normalization method of ultra high performance liquid chromatography (UHPLC)HPLC (Figure 8). 

For in vivo animal experiments, water solvent was used to suspend acteoside and the oral administration dosages were 20, 40, and 80 mg/kg/day for mice at a volume of 1 mL/100 g body weight; for in vitro cell studies, acteoside was dissolved in dimethyl sulfoxide (DMSO) and diluted with Dulbecco’s modified eagle media (DMEM) medium with a final concentration of 10^−7^, 10^−6^, and 10^−5^ mol/L.

### 4.2. Materials and Reagents

Acteoside was isolated from *C. deserticola* by using the above method (Figure 8, UHPLC ≥ 98%); RAW264.7 cells were obtained from Zhong Qiao Xin Zhou Biotechnology Co., Ltd. Shanghai, China; estradiol valerate was purchased from Delpharm Lille S.A.S, Paris, France; BGP, DPD, and TRAP crosslinks Elisa kits were bought from Xinyu Biological Engineering Co. Ltd., Shanghai, China; cathepsin K Elisa kit was purchased from Mountain View, CA, USA; MCSF and RANKL were provided by Pepro Tech Inc. American; total protein extraction kit and bicinchoninic acid (BCA) protein quantization kit were supplied by Ken Gen Biotech. Co. Ltd., Nanjing, China; primary antibodies including TRAF6, RANKL, RANK, NFKBIA, PI3K, AKT, IKKβ, NFAT2, c-Fos, β-actin, and secondary antibodies of horseradish peroxidase-conjugated goat anti-rabbit IgG were offered by ZSGB-BIO, Beijing, China; all other reagents were of analytical purity.

### 4.3. Laboratory Animals

All animal experiments were performed in accordance with the laboratory animal principles and approved on March 11, 2013 by the Bioethics Committee of the Ningxia Medical University (NXMU-20130311). Sixty female Kunming strain mice (8-week-old), provided by Laboratory Animal Centre of Ningxia Medical University with initial body weights of 26.0 ± 1.62 g, were housed in a regulated and standard specific pathogen-free environment (24.0 ± 0.5 °C, 45%–50% humidity and 12/12 h light/dark illumination cycles). All mice were allowed free access to tap water and diet. After being acclimated for 1 week, all mice were anesthetized by injecting 7% chloral hydrate (100 mg/kg, i.p.) under sterile conditions; then sham ovariectomy (SHAM), or two ovaries were all removed and randomly sub-divided into five groups (10 mice in each group) as follows: OVX (orally administrated with water) as model group, EV (1 mg/kg/day of estradiol valerate) as positive control; AceL, AceM and AceH (20, 40, 80 mg/kg/day of acteoside) as low-, moderate-, and high-dosage groups, respectively. 

All mice in the above six groups were orally administered daily and lasted for 12 weeks, with the dosage adjusted every two weeks according to body weight. At the end of the 3 months, the mice were euthanized and the serum (obtained by centrifuging whole blood at 4000 rpm for 15 min), uterus, vagina, right femora, and tibia were collected and stored at −80 °C for further analysis. 

### 4.4. Biomechanical Testing

Mice femoral biomechanical properties including maximal load and stiffness were measured by using a three-point bending testing with a BOSE machine (3220, Bose corporation, Endura TEC systems group, Minnetonka, MN, USA). Concisely, a V-shaped bending jig was required to secure the distal and the proximal ends of the femur during the measurement process, and the intact femur was placed horizontally with the front face up; then, the femur underwent a constant loading speed rate of 0.02 mm/s on two supports separated by a distance of 8 mm, and during the above process, the load and displacement curves were recorded until the femur was broken. From analyzing the load–displacement curves, the data of the maximal load and stiffness were obtained.

### 4.5. Bone Structure Analysis

Bone microarchitecture of the right distal femoral metaphysis was determined by using a micro-computer tomography (micro-CT) system (Locus SP, GE Healthcare, London, ON, Canada), with isotropic resolution of 8 µm in all three spatial dimensions. The region of interest was selected far away from the distal end of the femur, from 30 slices to 130 slices in the growth plate. The bone morphometric parameters including BMD, BMC, TMC, TMD, BVF, Tb. Sp, Tb. N, and Tb. Th were determined by analyzing the region of interest through using ABA bone analysis software.

### 4.6. Biochemical Parameter Analysis

The level of serum ALP was analyzed on an automatic analyzer (Ciba-Corning 550 Diagnostics, Corp., Oberlin, OH, USA). The content of serum DPD (Xinyu Biological Engineering Co. Ltd., Shanghai, China), and activities of TRAP (Xinyu Biological Engineering Co. Ltd.), cathepsin K (Mountain View, CA, USA), and BGP (Xinyu Biological Engineering Co. Ltd.) were measured by employing the corresponding Elisa reagent kits.

### 4.7. Protein Extraction and Western Blot Analysis

To induce mature osteoclast cells, RAW 264.7 cells were cultured by supplying with MCSF (30 ng/mL) and RANKL (25 ng/mL) for 6 days. The culture medium was replaced every 3 days. For cell survival test, RAW 264.7 cells were seeded in 96-well plates with the density of 2 × 10^5^ cells per well. After 6 days, different concentrations of acteoside (10^−5^, 10^−6^ and 10^−7^ mol/L, respectively) were added and then incubated at 37 °C in a humid atmosphere containing 5% CO_2_ for 24 and 48 h, respectively. Prior to the end of the culture, an MTT assay was used to detect the intervention effect of acteoside on osteoclast cells. For Western Blot analysis, the induced matured osteoclast cells were treated with acteoside (10^−5^, 10^−6^, and 10^−7^ mol/L, respectively) for 24 h, and then lysed by using a lysis buffer that contained 0.5 mmol phenylmethylsulfonyl fluoride, protease, and phosphatase inhibitors; next, the lysates were centrifuged and separated by employing 10% sodium dodecylsulfate-polyacrylamide gel electrophoresis (SDS-PAGE) and transferred to polyvinylidene difluoride (PVDF) membranes. After blocking with 5% skimmed milk for 60 min at ambient temperature, the membranes were probed with TRAF6 (1:500), RANKL (1:500), RANK (1:500), NF-κB (1:500), PI3K (1:500), AKT (1:500), NFAT2 (1:500), IKKβ (1:500), c-Fos (1:500), and β-actin (1:1000) antibodies, then the membranes were cultured overnight at 4 °C with these antibodies, followed by incubation with horseradish peroxidase-conjugated secondary antibody (1:2000); and finally, the blots were detected by the image lab software. The Western Blot experiments were repeated 3 times and β-actin was used as an internal control.

### 4.8. Statistical Analysis

All the results were expressed as mean ± SD. Statistical difference was identified by using SPSS (Version 18.0, IBM SPSS Statistics, IBM Corp., Armonk, New York, NY, USA) software with one-way analysis of variance and Dennett’s test method. A value of *p* < 0.05 indicated statistical significance. 

## 5. Conclusions

In conclusion, the present study revealed that acteoside exhibited potential activities for the prevention of bone loss in OVX mice, demonstrated by improving the trabecular microarchitecture and restoration of bone biomechanical competence. The underlying mechanism was mainly operated through the down-regulated RANKL/RANK/TRAF6 system, then the suppressed NF-κB and stimulated PI3K/AKT signaling pathways. These results indicate that acteoside could be a promising alternative replacement drug for the treatment of osteoporosis.

## Figures and Tables

**Figure 1 ijms-20-02974-f001:**
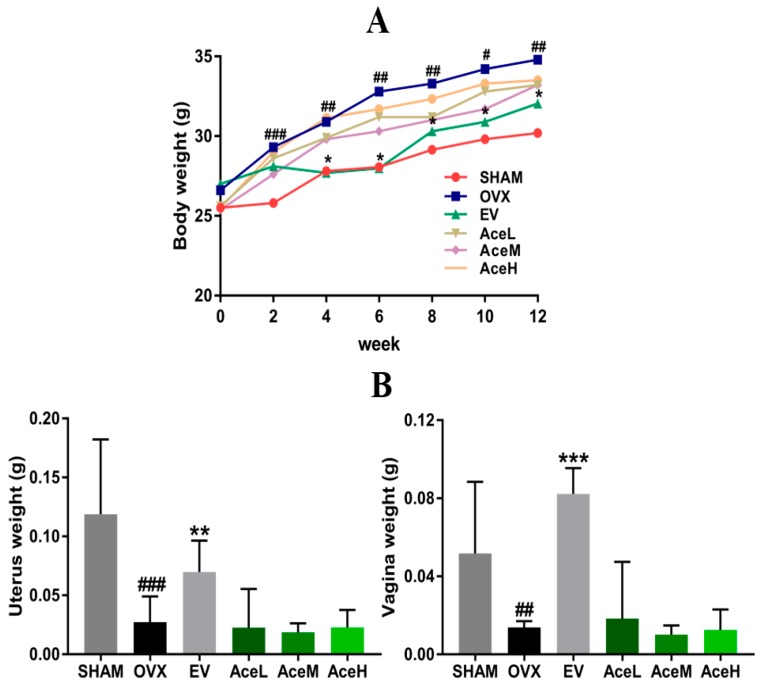
Effects of acteoside on body, uterine, and vaginal weights after being ovariectomized. (**A**) Body weight changes of all mice after surgery from 0 week to 12 weeks. (**B**) Effects of OVX (no treatment) and drug treatment on uterus and vagina wet weights of mice (*n* = 10); values were presented as the mean ± standard deviation (SD); ** *p* < 0.05, *** *p* < 0.001 relative to the mice of OVX model group; ^#^
*p* < 0.05, ^##^
*p* < 0.01, ^###^
*p* < 0.001 relative to the SHAM mice.

**Figure 2 ijms-20-02974-f002:**
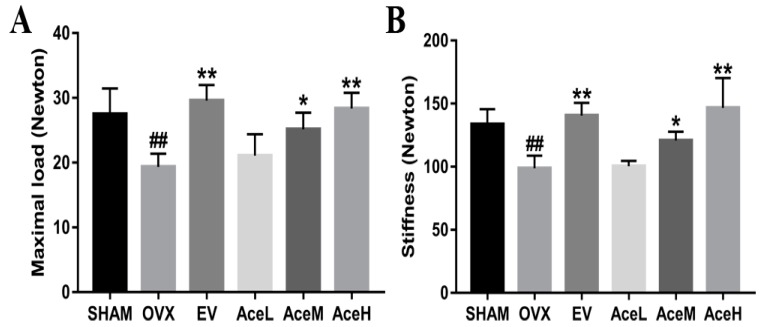
Effects of acteoside and EV on OVX-induced bone biomechanical changes of mice measured by using a three-point bending test (*n* = 4); (**A**) Maximal load. (**B**) Stiffness. Values are presented as the mean ± SD; * *p* < 0.05, ** *p* < 0.01 relative to the mice of OVX model group; ^##^
*p* < 0.01 relative to the SHAM mice.

**Figure 3 ijms-20-02974-f003:**
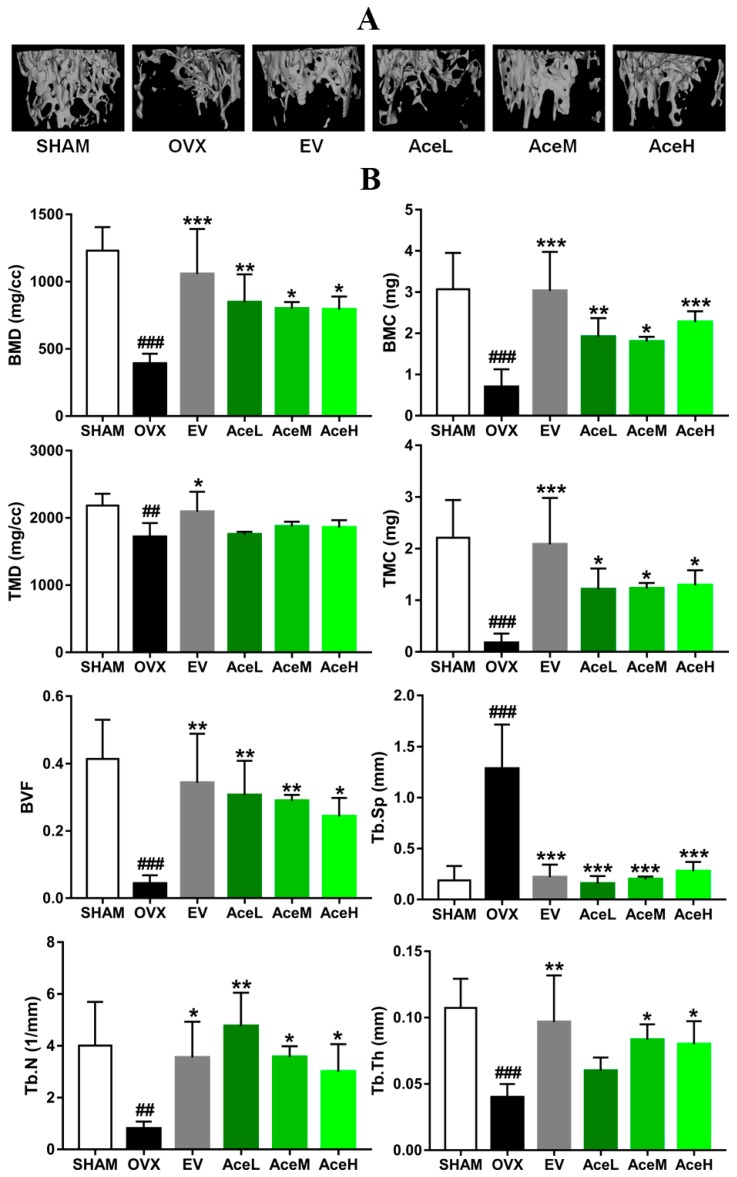
Acteoside treatments effectively prevented OVX-induced deterioration of trabecular microarchitecture in femur. (**A**) Right femur of mice from different groups were analyzed by micro-CT scans (*n* = 4), the trabecular area and trabecular number were significantly decreased in the OVX mice. After 12 weeks of treatment with acteoside and EV, the OVX-induced bone loss was partially prevented, and the microarchitecture was slightly improved. (**B**) Effects of acteoside and EV on OVX-induced microarchitecture changes. Microarchitecture parameters were described as bone mass density (BMD), bone mineral content (BMC), tissue mineral density (TMD), tissue mineral content (TMC), bone volume fraction (BVF), trabecular separation (Tb.Sp), trabecular number (Tb.N), and trabecular thickness mineral (Tb.Th) (*n* = 4). Values were presented as the mean ± SD; * *p* < 0.05, ** *p* < 0.01, *** *p* < 0.001 relative to the mice of OVX model group; ^##^
*p* < 0.01, ^###^
*p* < 0.001 relative to the SHAM group.

**Figure 4 ijms-20-02974-f004:**
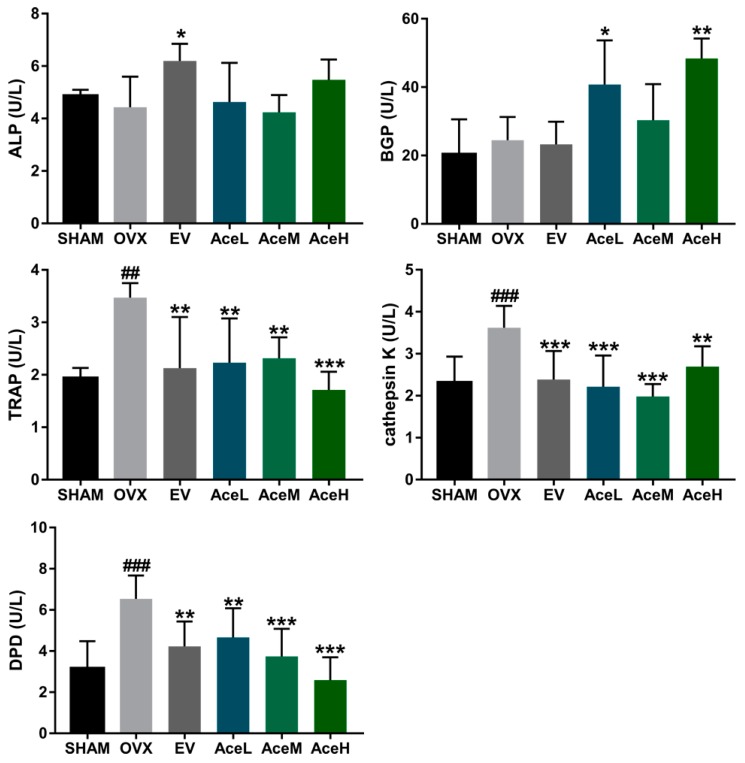
Acteoside treatment effectively prevented OVX-induced bone formation markers including ALP and BGP, and bone resorption indexes including TRAP, cathepsin K, and DPD activities (*n* = 10); values were presented as the mean ± SD. * *p* < 0.05, ** *p* < 0.01, *** *p* < 0.001 relative to the mice of OVX model group; ^##^
*p* < 0.001, ^###^
*p* < 0.001 relative to the SHAM group.

**Figure 5 ijms-20-02974-f005:**
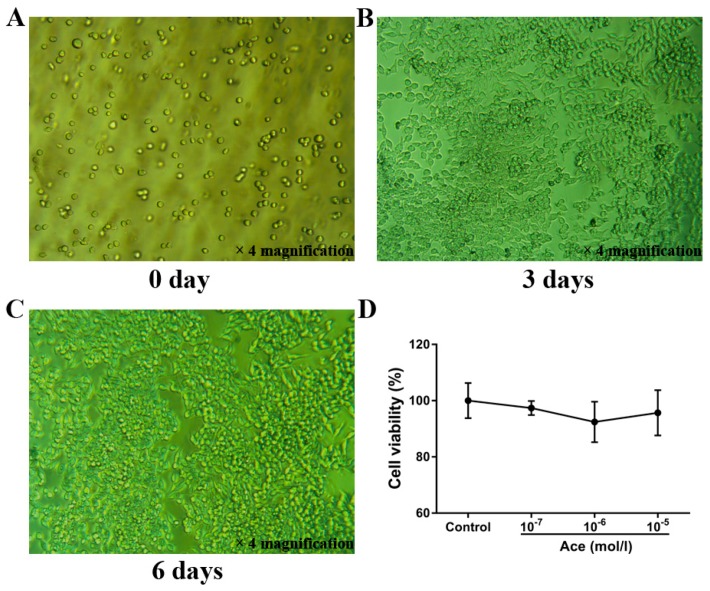
Acteoside treatment on RAW 264.7 cells. (**A**) RAW264.7 cells were incubated without MCSF (30 ng/mL) and RANKL (25 ng/mL). (**B**) RAW264.7 cells were incubated with MCSF (30 ng/mL) and RANKL (25 ng/mL) for 3 days (no treatment with acteoside). (**C**) RAW264.7 cells were incubated with MCSF (30 ng/mL) and RANKL (25 ng/mL) for 6 days (no treatment with acteoside). (**D**) Effects of acteoside on the cell viability of RAW264.7, cells were treated with the indicated concentrations of acteoside for the indicated times. Cell viability was determined by the MTT method.

**Figure 6 ijms-20-02974-f006:**
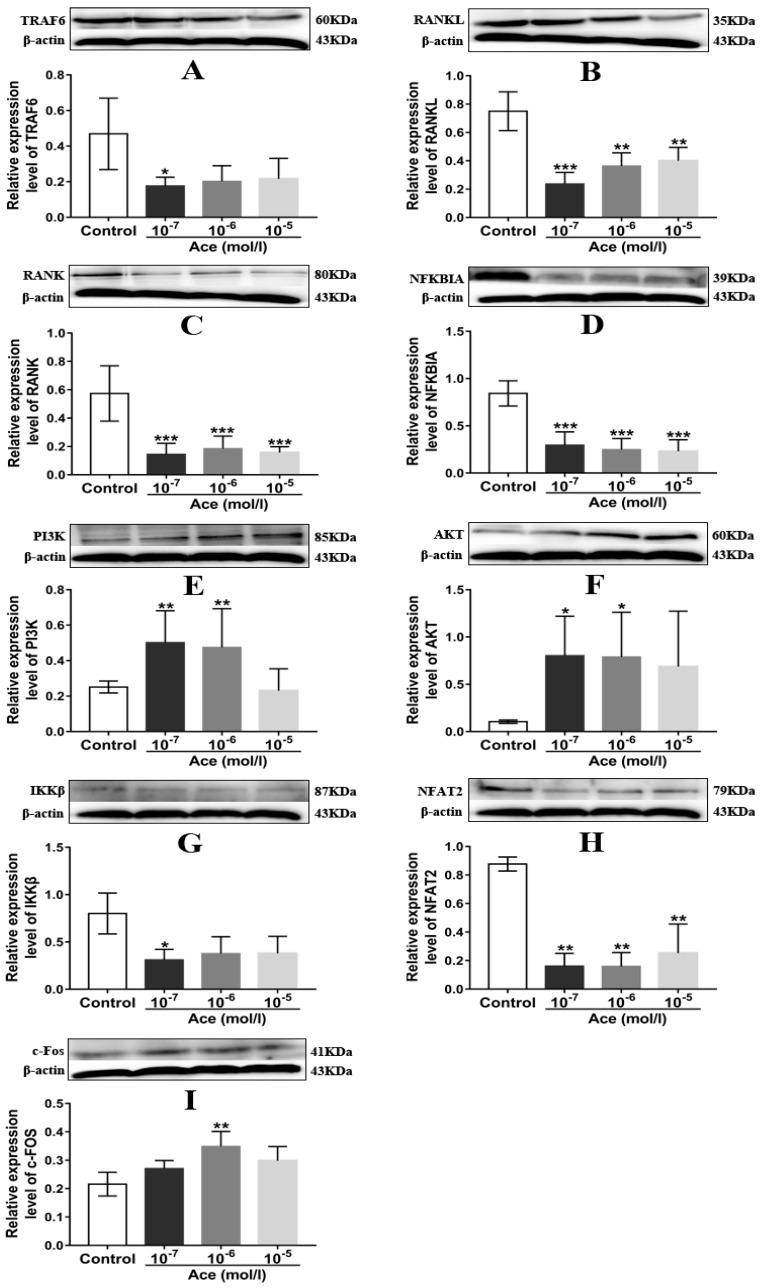
Acteoside treatment on the expression levels of TRAF6 (**A**), RANKL (**B**), RANK (**C**), NFKBIA (**D**), PI3K (**E**), AKT (**F**), IKKβ (**G**), NFAT2 (**H**), and c-Fos (**I**) (*n* = 3); β-actin was shown as the loading control, and quantitative data of every signal protein was descriptive as percentages of the value of control. Values were expressed as the mean ± SD. * *p* < 0.05, ** *p* < 0.01, *** *p* < 0.001 relative to the control group.

**Figure 7 ijms-20-02974-f007:**
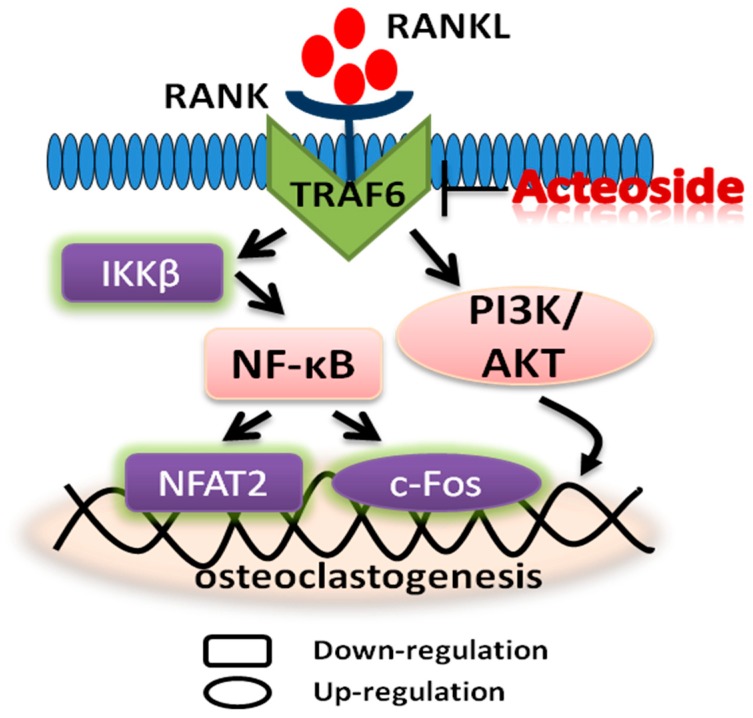
An overall hypothesis of the molecular mechanism: Acteoside could prevent bone loss on OVX mice through RANKL/RANK/TRAF6-mediated NF-κB inactivation and PI3K/AKT stimulation, which is shown by the protein levels of TRAF6, RANKL, RANK, NF-κB, IKKβ, and NFAT2 being down-regulated, while the expressions of PI3K, AKT, and c-Fos were up-regulated as compared to the control.

**Figure 8 ijms-20-02974-f008:**
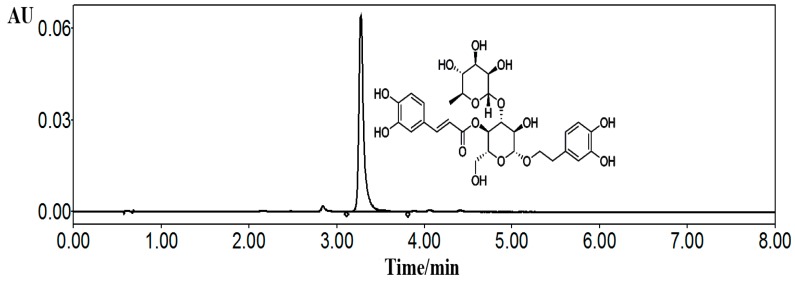
Chemical structure and ultra high performance liquid chromatography (UHPLC) chromatogram of acteoside. The chromatographic conditions were as follows: XBridge BEH C_18_ column (2.1 mm × 100 mm; 1.7 µm, Waters, Milford, MA, USA) at ambient temperature; mobile phase contained water with 0.5% v/v acetic acid (solvents A) and acetonitrile (solvents B), the gradient elution were: 0–2 min, 85–83% A; 2–5 min, 83–80% A; 5–8 min, 80–78% A; the flow rate was 0.4 mL/min and sample injection volume was 1 µL; the detection wavelength was 333 nm. Acteoside was dissolved in methanol for UHPLC analysis.

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
