# Peer review of "Protective Effect of Acteoside on Ovariectomy-Induced Bone Loss in Mice"

_ijms, 2019, doi:10.3390/ijms20122974_

Reviewer 1 Report

Osteoporosis displays a severe health burden and is expected to rise in prevalence. Therefore, it is essential to find effective and safe therapeutic strategies. In their study, Yang et al. aimed to elucidate the effect of supplementation of acteoside to mice upon ovariectomy. For this purpose, they have used a comprehensive approach by testing three different concentrations of acteoside and by performing µCT, biomechanical studies as well as molecular analyses. While the implications of the work are important, the results in their present form leave several questions. A number of major and minor points should be addressed prior to publication as detailed below.

The methods and results shown in Figure 5A are not clear. It is mentioned in the legend of that “RAW264.7 cells were incubated without or with MCSF (30 ng/ml) and RANKL (25 ng/ml) for 3 and 6 days and acteoside at the indicated concentrations.”. However, only three photos are shown (“0 day(4x)”, “3 days(4x)”, “6 days(40x)”). Are these the untreated or treated cells? Please show both. What was the used acteoside concentration? Please show the 40x magnification for all conditions and add a scale bar.

The methods and results are not well explained. For instance, it should be mentioned in this section what “AceH”, “AceM”, AceL” means and which acteoside were used in the respective conditions. Moreover, it should be explained why estradiol valerate was used as a control. The authors should help the readers (also the ones from different research fields) to understand their approaches.

Subdivide Figure 1 and Figure 2 in A, B, etc.

Please add error bars to Figure 5B.

The discussion is too long and unfocused. Please concentrate on aspects related to the current study.

There are serious language errors. A native speaker has to be consulted to revise the language.

To help the reader understand the purpose of the experiments, please start each result section by giving a short introduction (“To test whether […], we performed […]”).

Please define meanings of acronyms the first time they are being used.

Please italicize “in vivo” and “in vitro”.

Author Response

1. The methods and results shown in Figure 5A are not clear. It is mentioned in the legend of that “RAW264.7 cells were incubated without or with MCSF (30 ng/ml) and RANKL (25 ng/ml) for 3 and 6 days and acteoside at the indicated concentrations.”. However, only three photos are shown (“0 day(4x)”, “3 days(4x)”, “6 days(40x)”). Are these the untreated or treated cells? Please show both. What was the used acteoside concentration? Please show the 40x magnification for all conditions and add a scale bar.

Response:  Thank you for your careful and insightful comments; we now improve the resolution of Figure 5 and supply the untreated cells with 4x magnification in our revised manuscript.

2. The methods and results are not well explained. For instance, it should be mentioned in this section what “AceH”, “AceM”, AceL” means and which acteoside were used in the respective conditions. Moreover, it should be explained why estradiol valerate was used as a control. The authors should help the readers (also the ones from different research fields) to understand their approaches.

Response:  We improved the methods and results section of our manuscript, including explanation the meaning of “AceH, AceM, AceL” in “4. Materials and methods” section, explanation why estradiol valerate was used as positive control in “3. Discussion” section. In order to help readers to understand the methods, we also complement the aim of each experiment before the results in our revised manuscript.

3. Subdivide Figure 1 and Figure 2 in A, B, etc.

Response:  We now subdivide Figure 1 and Figure 2 in A, B section.

4. Please add error bars to Figure 5B.

Response: We now complement the error bars to Figure 5B.

5. The discussion is too long and unfocused. Please concentrate on aspects related to the current study.

Response: We have made an extensive revision and deleted the unfocused content in “3. Discussion” section of our manuscript.

6. There are serious language errors. A native speaker has to be consulted to revise the language.

Response: We invited a native English speaking colleague to help us checking grammar, spelling, punctuation and style of our manuscript.

7. To help the reader understand the purpose of the experiments, please start each result section by giving a short introduction (“To test whether […], we performed […]”).

Response: Thank you for your careful and insightful comments, and according to your suggestion, we complement the purpose of each experiment in each result section.

8. Please define meanings of acronyms the first time they are being used.

Response: We complemented the meaning of acronyms the first time they are being used in “Abstract” and “1. Introduction” section.

9. Please italicize “in vivo” and “in vitro”.

Response: We italicized “in vivo” and “in vitro” in our revised manuscript.

Reviewer 2 Report

The authors presented a study of the skeletal effects of acteoside on a model of bone loss due to ovariectomy. They demonstrated that acteoside could prevent skeletal degeneration marked by microstructural changes and serum biomarker in ovariectomized mice. They also showed that the mechanism of acteoside could be based on suppression of differentiation of osteoclasts. The methodology and results are sound but some major improvements in the presentation are needed. This manuscript also requires a major language revision (compulsory). There are many language errors and unstandardized expression.

Introduction: “…synthesized drugs always companied by kinds of adverse effect which made these agents so far from ideal” This is not an objective statement. Despite the side effects, the current anti-osteoporosis drugs are still effective in improving bone mineral density and preventing fracture.

Approval code from animal ethics committee not mentioned in the manuscript.

The resolution of figures embedded in the text is very poor. The reviewers cannot appreciate the details of the figure.

Figure 1-4: I suggest to rearrange the groups in the bar charts, from low to high acteoside, rather than high to low acteoside.

Figure 5: Why the magnification of micrographs on Day 0 (x4), 3 (x4) and 6 (x40) is different? The resolution is also very poor. Why the line graph doesn’t error bars?

Figure 6: The resolution of the charts is so poor that the reviewer cannot read the label of the x-axis. Is the concentration arranged from low to high or high to low? The reviewer suggests low to high as above.

Figure 7: The authors should indicate ‘Ace’ as acteoside.

Discussion: Content in paragraph 1 should appear in the Introduction.

Discussion: There is a lack of comparison of similar bone loss study using acteoside in the Discussion. The authors should also discuss the lack of dose-dependent effects for acteoside.

Discussion: Some of the limitations of study should be discussed: the lack for hormonal parameters, the use of young mice as a model (as opposed to using aged mice), the lack of cortical parameters for microCT, the limited biomechanical strength parameters (strength, elasticity and Young modulus), the lack of cellular histomorphometry indices.  

Author Response

Review #2:

1. Introduction: “…synthesized drugs always companied by kinds of adverse effect which made these agents so far from ideal” This is not an objective statement. Despite the side effects, the current anti-osteoporosis drugs are still effective in improving bone mineral density and preventing fracture.

Response: Thank you for your professional comments. Now, in our revised manuscript, we rewrite the sentence from “…synthesized drugs always companied by kinds of adverse effect which made these agents so far from ideal” to “synthesized drugs to treat this disease are indeed effective, however, various kinds of adverse effects which maybe related to these agents make them so far from ideal”.

2. Approval code from animal ethics committee not mentioned in the manuscript.

Response: We complemented the approval code in “4. Materials and methods” section of our revised manuscript.

3. The resolution of figures embedded in the text is very poor. The reviewers cannot appreciate the details of the figure.

Response: We found that the resolution of our figures in “PDF” form is lower than in “Word” style, the resolution of figures themselves are big enough for further publication. However, we also made improvements in all the figures to make them clearer.

4. Figure 1-4: I suggest to rearrange the groups in the bar charts, from low to high acteoside, rather than high to low acteoside.

Response: Thank you for your warm suggestion. We now rearrange the groups from low to high dosage in our revised manuscript.

5. Figure 5: Why the magnification of micrographs on Day 0 (x4), 3 (x4) and 6 (x40) is different? The resolution is also very poor. Why the line graph doesn’t error bars?

Response: We unified the magnification of micrographs on different days and improved the resolution of figure as well as complemented the error bars of the line.

6. Figure 6: The resolution of the charts is so poor that the reviewer cannot read the label of the x-axis. Is the concentration arranged from low to high or high to low? The reviewer suggests low to high as above.

Response: Thank you for your warm suggestion. We now improve the resolution of the charts and rearrange the groups from low to high dosage in our revised manuscript.

7. Figure 7: The authors should indicate ‘Ace’ as acteoside.

Response: We complete the meaning of “AceH, AceM, AceL” in “4. Materials and methods” section in our revised manuscript.

8. Discussion: Content in paragraph 1 should appear in the Introduction.

Response: Paragraph 1 in “3.Discussion” is related to the theory of TCM, now we complete some description in the “1. Introduction” section in our revised manuscript.

9. Discussion: There is a lack of comparison of similar bone loss study using acteoside in the Discussion. The authors should also discuss the lack of dose-dependent effects for acteoside.

Response: Thank you for your careful comments. We now complete the comparison of similar bone loss study using acteoside.

10. Figure 6: The resolution of the charts is so poor that the reviewer cannot read the label of the x-axis. Is the concentration arranged from low to high or high to low? The reviewer suggests low to high as above.

Response: We now improve the resolution of the charts and rearrange the groups from low to high dosage in our revised manuscript.

11. Discussion: Some of the limitations of study should be discussed: the lack for hormonal parameters, the use of young mice as a model (as opposed to using aged mice), the lack of cortical parameters for microCT, the limited biomechanical strength parameters (strength, elasticity and Young modulus), the lack of cellular histomorphometry indices. 

Response: Thank you for your professional and insightful comments. In our initial manuscript, the biomechanical strength parameters, including maximal load and stiffness, have been supplied, as well as the cortical parameters of micro-CT. And in our revised manuscript, we have complemented the cellular histomarphometry indices. In our further study, when we estimate the phytoestrogen effect of acteoside, we will consider the hormonal parameters. In addition, 8-week mice are mature and can be used to induce the postmenopausal osteoporosis.

Reviewer 3 Report

This manuscript is replete with serious grammatical errors and has very poor readability making it extremely difficult to evaluate the substantive content. The manuscript requires expert English language editing before resubmitting.

See attached pdf for some examples (not all) of language corrections and comments.

Author Response

Cite scientific references to back up this claim about TCM. Furthermore, to provide a more balanced view and avoid bias in this article, please acknowledge and, if possible, address the following issues:

1. Acteoside is a drug, and all drugs have adverse effects. What are the adverse drug effects of acteoside?

Response: Thank you for your insightful comments. Yet there is no published data revealed the adverse effect of acteoside, and as a promising anti-osteoporotic agent, we will estimate the toxicity of acteoside in the further study.

2. Osteoporosis is not caused by lack of drugs like acteoside. What are the causes of bone resorption in osteoporosis?

Response: Based on the data of our manuscript, we can see that during the process of postmenopausal osteoporosis, the activities of typical markers of bone resorption including TRAP, cathepsin K and DPD were significantly increased which maybe the causes of bone resorption in osteoporosis. However, since you raised this question, we found that the reasons of bone resorption in osteoporosis are not very clear in our head. We thank for your professional comments and we will consider them in our further research studies.

3. What are the potential problems with symptomatically treating bone resorption by suppressing osteoclastogenesis without removing the cause of osteoporosis? How is bone remodeling negatively impacted? (e.g. in osteonecrosis/osteopetrosis: https://www.ncbi.nlm.nih.gov/pubmed/16243172). Could these problems apply to acteoside? Are there long-term studies?

Response: Thank you for your professional comments. We indeed don’t know the above problems; the aim of our initial manuscript is just to estimate the anti-osteoporotic effect of acteoside, we have not mentioned the above problem. However, we feel that it is a long-term study to solve the problems which you mentioned, and the questions are very useful and we will consider them in our further research studies.

4. Regarding the role of the kidneys that you mentioned, please briefly summarize how the kidneys, bone, parathyroid glands, and intestines interact with hormones (fgf23, parathyroid hormone, vitamin D),  to regulate bone metabolism. How does acteoside interact with this regulatory system?

If you don't know the answers to these questions, please consider mentioning them as potential issues requiring further research studies involving acteoside.

Response: Thank you for your insightful comments. The TCM theory of “kidney dominates bone” is used to explain why we chose acteoside to estimate the anti-osteoporotic effect in our initial manuscript, however, we will consider your suggestions carefully and make a further study which mention the kidneys, bone, parathyroid glands and intestines interact with hormones to regulate bone metabolism. Thank your again for your professional and warm suggestions.

Round  2

Reviewer 1 Report

I thank the authors for the adequate adjustments.

The manuscript has significantly gained quality.
In my opinion, it is suitable for publication in "IJMS".

Author Response

Dear Editors and reviewers,

Enclosed is a revised manuscript entitled “Protective Effect of Acteoside on Ovariectomy-Induced Bone Loss in Mice” (ijms-509030 V2). Thank you very much for your suggestions on our manuscript. We also thank the reviewers for their careful review and insightful comments. According to the editors’ and reviewers’ comments, we have made a careful and major correction in our manuscript. All the changes we made in the manuscript were highlighted in red. Thank you very much for your consideration.

Sincerely,

Xueqin Ma

Department of Pharmaceutical,

School of Pharmacy,

Ningxia Medical University,

1160 Shenli Street

Yingchuan 750004, P.R.China

Tel& Fax: +86 951 6880693;

E-mailmaxueqin217@126.com

02/06/2019

Reviewer 2 Report

Thank you for the revised version. 

Line 126-127: the authors indicated that "micro-CT was usually used to assess bone fragility in clinical". This statement is not accurate because micro-CT is seldom used clinically and it is used to assess bone microarchitecture, not fragility. 

Figure 7: the 'T' arrow showing Acteocide inhibiting TRAF6 should be turned 180 degrees.

Line 256-259: There is no doubt that microCT provides more accurate measurement of bone structural parameters, but bone cellular parameters (osteoblast number, osteoclast number, osteoid surface) and bone dynamic parameters (mineralizing surface, bone formation rate) can only be determined using bone histomorphometrical methods. If the author did not have these data, please list them as limitations.  

Line 344: Please add "Approval code:" before "2013-137"

Despite the best attempt to improve the language of the manuscript, there are still many grammatical mistakes and confusing sentences in the text. Please kindly send the manuscript to be edited professionally.

Author Response

Dear Editors and reviewers,

Enclosed is a revised manuscript entitled Protective Effect of Acteoside on Ovariectomy-Induced Bone Loss in Mice” (ijms-509030 V2). Thank you very much for your suggestions on our manuscript. We also thank the reviewers for their careful review and insightful comments. According to the editors’ and reviewers’ comments, we have made a careful and major correction in our manuscript. All the changes we made in the manuscript were highlighted in red. What follows are the responses to the editors and reviewers:

Review #1:

1. Line 126-127: the authors indicated   that "micro-CT was usually used to assess bone fragility in   clinical". This statement is not accurate because micro-CT is seldom   used clinically and it is used to assess bone microarchitecture, not   fragility.

Response:  Thank you for your professional and   insightful comments; we now revise this sentence to make it more objective in   our revised manuscript.

2. Figure 7: the 'T' arrow showing   Acteocide inhibiting TRAF6 should be turned 180 degrees.

Response:  Thank you for your careful suggestion, we   now correct it in our revised manuscript.

3. Line 256-259: There is no doubt that   microCT provides more accurate measurement of bone structural parameters, but   bone cellular parameters (osteoblast number, osteoclast number, osteoid   surface) and bone dynamic parameters (mineralizing surface, bone formation   rate) can only be determined using bone histomorphometrical methods. If the   author did not have these data, please list them as limitations. 

Response:  Thank you for your professional suggestion.   We now list the above description as limitations in the section of “3.   Discussion” in our revised manuscript.

4. Line 344: Please add "Approval   code:" before "2013-137"

Response:  We now complement the approval code in our revised   manuscript.

5. Despite the best attempt to improve   the language of the manuscript, there are still many grammatical mistakes and   confusing sentences in the text. Please kindly send the manuscript to be   edited professionally.

Response:  Thank you for your warm suggestion. We have   made an extensive and careful revision in our revised manuscript. Besides   invited a native English speaking colleague to help us improving the   language, we also scrutinized all of the grammar, spelling and style one by   one, and we found that the “Track Changes” mode in Microsoft Word usually   make some errors which was not easy to detect. In order to illustrate our   revised manuscript clearly, all the changes we made in the manuscript were   highlighted in red instead of by using “Track Changes” function.

All the changes we made in the manuscript were highlighted in red. We believe that the revised manuscript is in good shape after revision according to the reviewer’s and editor’s suggestions. Please consider for acceptance to be published in “International Journal of Mechanical Sciences” and let us know any further information of the reviewing process at your earliest convenience.

Thank you very much for your consideration.

Sincerely,

Xueqin Ma

Department of Pharmaceutical,

School of Pharmacy,

Ningxia Medical University,

1160 Shenli Street

Yingchuan 750004, P.R.China

Tel& Fax: +86 951 6880693;

E-mailmaxueqin217@126.com

02/06/2019

Round  3

Reviewer 2 Report

Thank you to the authors for addressing my comments.